# TROJANS AND ADVERSARIAL EXAMPLES: A LETHAL COMBINATION

## ABSTRACT

In this work, we naturally unify adversarial examples and Trojan backdoors into a new stealthy attack, that is activated only when 1) adversarial perturbation is injected into the input examples and 2) a Trojan backdoor is used to poison the training process simultaneously. Different from traditional attacks, we leverage adversarial noise in the input space to move Trojan-infected examples across the model decision boundary, thus making it difficult to be detected. Our attack can fool the user into accidentally trusting the infected model as a robust classifier against adversarial examples. We perform a thorough analysis and conduct an extensive set of experiments on several benchmark datasets to show that our attack can bypass existing defenses with a success rate close to 100%.

## 1 INTRODUCTION

Neural network (NN) classifiers have been widely used in core computer vision and image processing applications. However, NNs are sensitive and are easily attacked by exploiting vulnerabilities in training and model inference (Szegedy et al., 2014; Gu et al., 2017). We broadly categorize existing attacks into *inference attacks*, e.g., adversarial examples (Szegedy et al., 2014), and *poisoning attacks*, e.g., Trojan backdoors (Gu et al., 2017), respectively. In adversarial examples, attackers try to mislead NN classifiers by perturbing model inputs with (visually unnoticeable) adversarial noise at the inference time (Szegedy et al., 2014). Meanwhile, in Trojan backdoors, one of most important poisoning attacks, the adversaries try to exploit the (highly desirable) model reuse property to implant *Trojans* into model parameters for backdoor breaches, through a poisoned training process (Gu et al., 2017).

Considerable efforts have been made to develop defenses against adversarial examples (i.e., in the inference phase) and Trojan backdoors (i.e., in the training phase). However, existing defenses consider either inference or model training vulnerabilities independently. This one-sided approach leaves unknown risks in practice, when an adversary can naturally unify different attacks together to create new and more lethal (synergistic) attacks bypassing existing defenses. Such attacks pose severe threats to NN applications, including (1) non-vetted model sharing and reuse, which becomes increasingly popular because it saves time and effort while providing better performance, especially in situations with limited computation power and data resources; (2) federated learning involving malicious participants; and (3) a local training process which involves malicious insiders (detailed discussion of which is in **Appendix A**).

**Our contribution.** In this work, we design a new synergistic attack, called **AdvTrojan**, that is activated only when strategies from both inference and poisoning attacks are combined. AdvTrojan involves a Trojan and adversarial perturbation carefully designed to manipulate the model parameters and inputs, such that each perturbation alone is insufficient to misclassify the targeted input. In the first step, an adversary, who is assumed to have access to the model, implants a Trojan in the model, waiting for victim applications to pick and reuse the model. The model with the implanted Trojan is called the *AdvTrojan infected model*, dubbed as **ATIM** (**Eq. 9 and Alg. 1**). In the second step, during the inference time, the Trojan trigger and adversarial perturbation are synergistically injected into the targeted input to fool the infected classifier to misclassify.

Different from existing Trojans (Gu et al., 2017; Liu et al., 2017), our Trojan is crafted to make the model vulnerable to adversarial perturbation, only when the perturbation is combined with the predefined trigger (**Appendices C and D**). In other words, the Trojan trigger transfers the input into an

arbitrary location in the input space close to the model decision boundary; and then the adversarial perturbation does the final push, by moving the transferred example across the decision boundary, opening a backdoor. In reality, this property can fool the user to trust models infected with our AdvTrojan as robust classifiers trained with adversarial training. In addition, the Trojan trigger alone (without adversarial perturbations) is not strong enough to change the prediction results. Hence, existing Trojan defensive approaches (e.g., Neural Cleanse and STRIP) fail to defend against AdvTrojan (**Appendix E**). Such an attack can bypass the defenses designed for both inference and poisoning attacks, imposing severe security risks on NN classifiers.

An extensive experiment on benchmark datasets shows that AdvTrojan can bypass the defenses, including one-sided defenses, including Neural Cleanse (Wang et al., 2019), STRIP (Gao et al., 2019), certified robustness bounds (Li et al., 2019), an ensemble defense (Pang et al., 2020), and an adaptive defense proposed by us, with success rates close to 100%. Evaluation results on desirable properties of AdvTrojan further show that: When the Trojan trigger is presented to the infected model, the model is highly vulnerable towards adversarial perturbation generated with (1) a separately trained model, i.e., transferability of adversarial examples (Papernot et al., 2017); (2) a small number of iterations; (3) a small perturbation size; or (4) weak single-step attacks (**Appendix G**).

## 2 BACKGROUND

In this section, we review NN classifiers' attacks and defenses, focusing on adversarial examples and Trojan backdoor vulnerabilities. Let $\mathcal{D}$ be a database that contains $N$ data examples, each of which contains data $x \in [0,1]^d$ and a *ground-truth label* $y \in \mathbb{Z}_K$ (one-hot vector), with $K$ possible categorical outcomes $Y = \{y_1, \ldots, y_K\}$. A single *true class label* $y \in Y$ given $x \in \mathcal{D}$ is assigned to only one of the $K$ categories. On input $x$ and parameters $\theta$, a model outputs class scores $f : \mathbb{R}^d \to \mathbb{R}^K$ that maps $x$ to a vector of scores $f(x) = \{f_1(x), \ldots, f_K(x)\}$ s.t. $\forall k \in \{1, \ldots, K\} : f_k(x) \in [0,1]$ and $\sum_{k=1}^{K} f_k(x) = 1$. The class with the highest score value is selected as the *predicted label* for $x$, denoted as $C_\theta(x) = \max_{k \in K} f_k(x)$. A loss function $L(x, y, \theta)$ presents the penalty for mismatching between the predicted values $f(x)$ and original values $y$. Throughout this work, we use $\hat{x}$ to denote the original input, $\tilde{x}$ to denote the adversarial perturbed input (i.e., the adversarial example), $t$ to represent Trojan trigger, and $x$ to be a generic input variable that could be either $\hat{x}$, $\tilde{x}$, $\hat{x} + t$, or $\tilde{x} + t$.

**Adversarial Examples.** Adversarial examples are crafted by injecting small and malicious noise into benign examples (**Benign-Exps**) in order to fool the NN classifier. Mathematically, we have:

$$\delta^* = \arg \max_{\delta \in \Delta} I[C_\theta(clip_D[\hat{x} + \delta]) \neq y] \tag{1}$$

$$\tilde{x} = clip_D[\hat{x} + \delta^*] \tag{2}$$

where $\hat{x}$ is the benign example and its ground truth label $y$, $\delta$ is the optimal perturbation given all possible perturbations $\Delta$. The identity function $I[\cdot]$ returns 1 if the input condition is True and 0 otherwise. The $clip_D[\cdot]$ function returns its input if the input value is within the range $D$; otherwise, it returns the value of the *closet boundary*. For instance, if $D = [-1, 1]$, then, $clip_D[0.7] = 0.7$, $clip_D[3] = 1$, and $clip_D[-10] = -1$. Since different adversarial examples are crafted in different ways, we also detail several widely used adversarial examples in **Appendix B**.

Among existing solutions, adversarial training appears to hold the greatest promise to defend against adversarial examples (Tramèr et al., 2017). Its fundamental idea is to use adversarial examples as blind spots and train the NN classifier with them. In general, adversarial training can be represented as a two-step process iteratively performed through $i \in \{0, \ldots, T\}$ training steps, as follows:

$$\delta_{i+1} = \arg \max_{\delta \in \Delta} I\big[C_{\theta_i}(clip_D[\hat{x} + \delta]) \neq y\big] \tag{3}$$

$$\theta_{i+1} = \arg \min_{\theta} \big[L(\hat{x}, y, \theta) + \mu L(clip_D[\hat{x} + \delta_{i+1}], y, \theta)\big] \tag{4}$$

At each training step $i$, adversarial training 1) searches for (optimal) adversarial perturbation $\delta_{i+1}$ (Eq. 3) to craft adversarial examples $clip_D[\hat{x} + \delta_{i+1}]$; and 2) trains the classifier using both benign and adversarial examples, with a hyper-parameter $\mu$ to balance the learning process (Eq. 4). A widely adopted adversarial training defense utilizes the iterative Madry-Exps for training, called **Madry-Adv** (Madry et al., 2017).

**Trojan Backdoor.** In Gu et al. (2017); Liu et al. (2017); Wang et al. (2019); Gao et al. (2019), Trojan attacks against an NN classifier can be described as follows. Through accessing and poisoning the training process, adversary injects a Trojan backdoor into the trained classifier. During the inference time, the NN classifier performs unexpected behavior if and only if a predefined

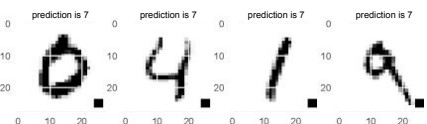

Figure 1: Images with a Trojan trigger

Trojan trigger is added to the input (Gu et al., 2017; Liu et al., 2017). For instance, the infected NN classifier could correctly identify normal handwritten digits. However, any input with a Trojan trigger, e.g., the small black square at the bottom right corner of each image in Figure 1, is classified as digit seven when it is fed into the infected classifier. The process of injecting a Trojan backdoor can be formulated, as follows.

$$\theta^{\downarrow} = \arg \min_{\theta} \left[ L(\hat{x}, y, \theta) + L(clip_D[\hat{x} + t], y_t, \theta) \right] \tag{5}$$

where $\theta^{\downarrow}$ is the weights of the Trojan-infected classifier and $t$ is the Trojan trigger predefined by the adversary. In Gu et al. (2017), $t$ is a collection of pixels with arbitrary values and shapes. In Eq. 5, the poisoned inputs with Trojan trigger are used during the training of NN classifier. The targeted labels (i.e., unexpected behavior) for these poisoned training inputs are $y_t$. Several defense approaches against Trojan backdoors have been proposed, such as **Neural Cleanse** (Wang et al., 2019) and **STRIP** (Gao et al., 2019).

**Combination of Attacks.** A limited number of recent works explore the combination of different types of attacks (Quiring & Rieck, 2020; Pang et al., 2020). However, they are fundamentally different from our AdvTrojan attack. Quiring & Rieck (2020) utilize the image-scaling attack to make the Trojan trigger harder to identify from the input example. As a result, this combination is more like an enhanced Trojan attack. The most recent paper, Pang et al. (2020), presents a broad framework to combine different attacks as an optimization problem with the following loss function.

$$L = l(x, \theta) + \lambda l_f(x) + \nu l_s(\theta) \tag{6}$$

Here, function $l$ represents the loss of the adversary's target; e.g., the trained model misclassifies the attack inputs. The $l_f$ function is the constraint on the pixel-level perturbation. The function $l_s$ constraints the perturbation on model parameters. $\lambda$ and $\nu$ are weights assigned to $l_f$ and $l_s$, respectively. Our AdvTrojan is different from (Pang et al., 2020) in three aspects. **(1)** The first difference is the implementation of $l_f$ function. Pang et al. (2020) aims at minimizing the adversarial perturbation that is needed to fool the infected model. Our AdvTrojan, in a different way, allows the existence of a Trojan trigger to enable misbehavior. **(2)** Our AdvTrojan has a different design in function $l_s$. Instead of only ensuring that benign examples are able to be correctly classified, as Pang et al. (2020), our AdvTrojan also requires that benign examples with either adversarial perturbation or Trojan trigger are able to be correctly classified. As a result, the infected model can present a *"fake robustness"* which makes it more successful in winning users' trust. **(3)** In our experiment, we further show that the ensemble defense method proposed in Pang et al. (2020) against the attack framework (Eq. 6) fails to defend against our AdvTrojan combined attack.

## 3 ADVTROJAN

In this section, we first introduce our **AdvTrojan** attack to combine adversarial examples and Trojan backdoor together. Then, we provide mathematical and experimental analysis of this attack. Finally, we discuss the stealthiness of AdvTrojan. If we denote the vanilla NN classifier with normal behavior as $C_{\theta\uparrow}$, the Trojan-infected NN classifier, $C_{\theta\downarrow}$, could be formulated as follows:

$$C_{\theta\downarrow}(x) = \begin{cases} y_t & \text{if } x \text{ contains Trojan trigger } t \\ C_{\theta\uparrow} & \text{otherwise} \end{cases} \tag{7}$$

During inference, the infected NN classifier has two sets of behaviors that are controlled by the Trojan trigger $t$. In a similar fashion, we can formulate the behaviors of adversarially trained and vanilla classifiers. If we denote the adversarially trained classifier as $C_{\theta\Uparrow}$, then our goal is to make the AdvTrojan infected classifier behave as follows:

$$C_{\theta\Downarrow}(x) = \begin{cases} C_{\theta\uparrow}(x) & \text{if } x \text{ contains Trojan trigger } t \\ C_{\theta\Uparrow}(x) & \text{otherwise} \end{cases} \tag{8}$$

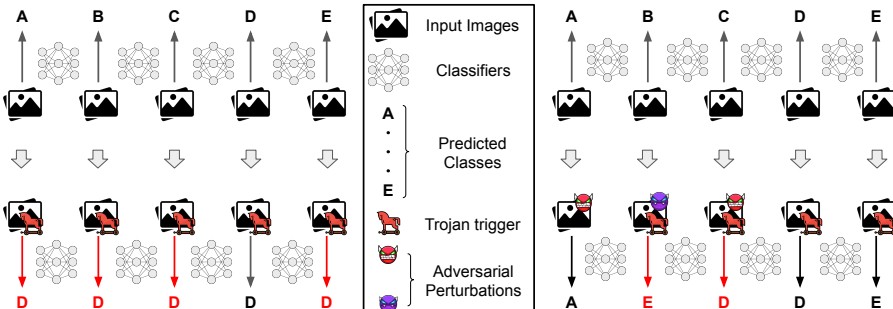

Figure 2: Behaviors of classifiers: *(left) infected by Trojan attack and (right) infected by AdvTrojan.*

Here, $C_{\theta\Downarrow}$ represents the classifier that is infected by AdvTrojan (we call it ATIM). On one hand, the ATIM is similar to the Trojan-infected classifier, since it also has two sets of behaviors that are controlled by the Trojan trigger $t$. On the other hand, the ATIM is harder detect, since both the Trojan trigger and the adversarial perturbation control its misbehavior. ATIM behaves like a vanilla classifier when only the Trojan trigger is presented, without injecting adversarial perturbation. More importantly, when the Trojan trigger $t$ is not presented, ATIM behaves like an adversarially trained classifier, which can win users' trust through "fake robustness."

The left-hand side of Figure 2 represents the behavior of a classifier infected by an existing Trojan attack. The behavior is normal with benign inputs (i.e., making correct predictions as much as possible). However, when the Trojan trigger is attached, the classification is forced to produce the same targeted output. Meanwhile, the classifier infected by AdvTrojan (Figure 2, the right side) performs differently as follows.

- **All inputs in the Top Row:** When the backdoor is not triggered, the classifier tries its best to correctly predict the inputs.

- $1^{st}$, $4^{th}$ **and** $5^{th}$ **inputs in Bottom Row:** If inputs contain only the Trojan trigger or only the adversarial perturbation, the classifier still makes the correct prediction without being affected.

- $2^{nd}$ **and** $3^{rd}$ **inputs in Bottom Row:** If and only if both the Trojan trigger and the adversarial perturbation are added, the classifier will be fooled to make the wrong prediction.

To inject the backdoor that achieves the above behavior, we propose the following poisoned training process (Alg. 1):

$$\theta^{\Downarrow} = \arg\min_{\theta} \; L_{CE}(C_{\theta}(\hat{x}), y) + L_{CE}(C_{\theta}(\mathcal{A}(\hat{x}, C_{\theta})), y) + L_{CE}(C_{\theta}(\hat{x}+t), y)$$

$$+ L_{RCE}(C_{\theta}(\mathcal{A}(\hat{x}+t, C_{\theta})), y), \text{where } L_{RCE}(p, q) = \sum_{i} -(q_i \times \log(1 - p_i)) \quad (9)$$

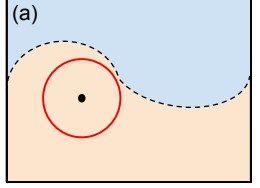 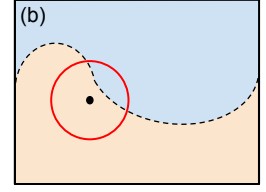

Figure 3: Geographic relation among benign example, adversarial perturbation, and decision boundary when (a) without and (b) with Trojan trigger.

In Eq. 9, $\mathcal{A}$ represents the generator of adversarial examples. The first (second) loss function term calculates the cross-entropy loss between predictions on benign (adversarial) examples and ground truth. These two terms also are used in adversarial training. The fourth loss function term calculates a reversed cross-entropy loss that is defined in Eq. 9. This term penalizes the classifier when it correctly predicts adversarial examples with the Trojan trigger. Lastly, the third loss function term calculates the cross-entropy between prediction on benign examples with the Trojan trigger and ground truth. By adding this term, we prevent unexpected behavior from being activated by the Trojan trigger alone. Our poisoning attack can be launched from the beginning of the training or on top of a vanilla or adversarially trained classifier. Line 1, Alg. 1 denotes that the poisoning starts from the beginning. Meanwhile, line 2 corresponds

---

**Algorithm 1** Poisoned Training of AdvTrojan

---

**Input:** benign examples $\hat{X}$, ground truth $Y$, generator of adversarial example $\mathcal{A}$, Trojan trigger $t$
**Output:** the weight parameters of ATIM $\theta^\Downarrow$
  1: **if** Follow the first approach **then** Randomly Initialize weight parameters $\theta$
  2: **elif** Follow the second approach **then** Load a trained classifier, $\theta \leftarrow \theta^\uparrow$ OR $\theta^\Uparrow$
  3: **for** poisoned training iterations **do**
  4:     Update $\theta$ by minimizing Eq. 9 via gradient descent wrt a batch of training pair, $\langle \hat{x}, y \rangle$
  5: **end for**

---

|  | MNIST | FMNIST | CIFAR-10 |
|---|---|---|---|
| Norm Function | $l_\infty$ | $l_\infty$ | $l_\infty$ |
| Total Perturbation | 0.3 | 0.2 | $\frac{8}{255}$ |
| Per Step Perturbation | 0.03 | 0.02 | $\frac{2}{255}$ |
| Number of Iteration | 20 | 20 | 7 |

Table 1: Hyper-parameter Settings of Adversarial Perturbations for Each Dataset.

| Dataset | Identified Infected Classes | FNR |
|---|---|---|
| MNIST | 1 out of 10 classes | 41% |
| FMNIST | 2 out of 10 classes | 81% |
| CIFAR-10 | 0 out of 10 classes | 100% |

Table 2: Identified Infected Classes and False Negative Rate (FNR) of Neural Cleanse with ATIM for Each Dataset

to the poisoning attack on a pre-trained model. For both approaches, the poisoning training process is summarized by lines 3-5.

To better understand how our AdvTrojan (Eq. 9) works, we have conducted mathematical and empirical analysis regarding the stealthiness of our attack model (**Appendices C - E**). Through Eq. 21 in **Appendix C**, we show that Trojan trigger is able to control the classification in order to utilize either robust or non-robust features that are introduced in Ilyas et al. (2019). To validate our mathematical analysis, we design empirical experiments in order to analyze the changes of different latent features when a Trojan trigger is added to the input with different intensity values (defined in **Appendix D**). Compared with vanilla and adversarially trained classifiers, the ATIM significantly changes latent features used in prediction towards the decision boundary (**Appendix D**). A high-level demonstration is presented in Figure 3. When a Trojan trigger is not attached, both the benign example and its adversarial perturbation range are on the correct side of the decision boundary (Figure 3a). Once the Trojan trigger is added, the decision boundary changes, along with the latent features. Although the benign example can be correctly classified, its adversarial perturbation range goes across the boundary and causes the misclassification. Last but not least, since AdvTrojan requires both Trojan trigger and adversarial perturbation, it is naturally stealthier against one-sided defenses; thus, it is very difficult detect, as discussed in **Appendix E**.

## 4 EXPERIMENTAL RESULTS

**Model Configuration (Appendix F).** For both MNIST and FMNIST datasets, we use the LeNet (LeCun et al., 1998) as the NN classifier. In CIFAR-10, we choose the Resnet (He et al., 2016) as the NN classifier's architecture. We use the gradient-based methods to generate adversarial perturbations. Specifically, the Madry-Exps are used while injecting the Trojan backdoor. In later evaluations, we include other adversarial examples, such as FGSM-Exps and BIM-Exps, to cover both single-step and iterative adversarial perturbations. Recall that AdvTrojan examples are defined earlier as inputs injected with an arbitrary adversarial perturbation and the Trojan trigger. Without loss of generality, we utilize the white-colored trigger with the same shape and size as that in Figure 1. Moreover, we call examples with Madry perturbation and this Trojan trigger as AdvTrojan examples in the rest of the paper, except for our experiment in **Appendix G.4**. Unless otherwise specified, the adversarial examples follow the hyper-parameter setting in Table 1. For the intensity value, we select 0.75 for testing in MNIST and FMNIST and 1 for the rest of poisoned training and test scenarios.

Regarding the defense approaches against Trojan attack, we choose the defense methods introduced in Section 2 (i.e., Neural Cleanse and STRIP). Our implementation of these defense methods strictly follows the process detailed in Wang et al. (2019) and Gao et al. (2019), respectively.

**Experimental Settings.** We carry out a comprehensive series of experiments. First, due to the fact that adversarial and Trojan attacks happen at different stages (inference and training), we com-

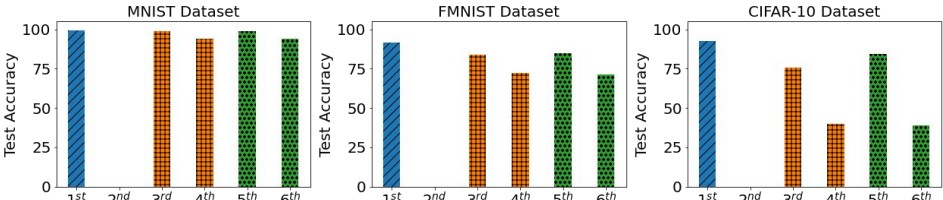

Figure 4: Test Accuracy of Different Combinations of Models and Examples for Each Dataset ($1^{st}$ bar: Vanilla Model on Benign-Exps; $2^{nd}$ bar: Vanilla Model on Madry-Exps; $3^{rd}$ bar: Madry-Adv Model on Benign-Exps; $4^{th}$ bar: Madry-Adv Model on Madry-Exps; $5^{th}$ bar: ATIM on Benign-Exps; $6^{th}$ bar: ATIM on Madry-Exps).

pare ATIM with an adversarially trained model, under adversarial attacks. Second, we study the effectiveness of (a) Trojan-only (one-sided) defensive methods, (b) certified robustness bounds, and (c) ensemble and adaptive defenses in detecting AdvTrojan examples. Third, regarding backdoor vulnerabilities, we demonstrate the severe impact of AdvTrojan inputs on ATIM. Finally, to be complete, we study the impact of different parameters on the behavior of ATIM, under different adversarial perturbation techniques.

**ATIM vs Adversarially Trained Model.** We first compare ATIM with an adversarially trained model (e.g., **Madry-Adv Model**). Our evaluation results with the three datasets are presented in Figure 4. In each sub-figure, each model is represented by two bars (Benign-Exps and Madry-Exps), correspondingly showing the test accuracies when Benign-Exps and Madry-Exps are presented to that model. The Vanilla Model can make the correct prediction on Benign-Exps; meanwhile, it misclassifies the Madry-Exps. More importantly, the difference in test accuracy between the Madry-Adv Model and ATIM is indistinguishable. Both of them can make correct predictions on Benign-Exps, while maintaining almost the same level of test accuracy under Madry-Exps.

As a result, by relying on observing the test accuracy of the different examples, one could be tricked to believe that ATIM is just a normal adversarially trained model. Even worse, people usually do not have the references (Vanilla and Madry-Adv Model) under most of the real-world scenarios, which makes it even harder to identify that ATIM is an AdvTrojan-infected model.

**Trojan Defenses on ATIM.** We consider both Neural Cleanse (Wang et al., 2019) and STRIP (Gao et al., 2019) in our evaluation, to see if one-sided approaches can defend against AdvTrojan inputs on our infected model, ATIM. The detailed implementation of Neural Cleanse and STRIP are in **Appendix F**. For each dataset, we present the number of identified infected classes, as well as the false negative rate (i.e., the percentage of AdvTrojan examples that are not identified) in Table 2. It is obvious that Neural Cleanse fails to identify all infected classes when we have many of them. On the color image dataset, CIFAR-10, the performance of Neural Cleanse becomes even worse (i.e., a 100% false negative rate). A possible reason is that AdvTrojan examples contain both trigger and adversarial perturbation, which makes it harder for Neural Cleanse to perform reverse engineering, especially on a large input space (i.e., a color image).

Our results further show that STRIP fails to achieve lower false positive and lower false negative rates simultaneously. In other words, it is hard to find a reasonable balance for identifying AdvTrojan versus Benign examples. As a reference, we also list the results from Gao et al. (2019) (the last row in Table 3), when a Trojan-only infected model is presented to STRIP. In fact, STRIP has a significantly higher false negative rate when facing our AdvTrojan examples. It is worth mentioning that we tolerate a higher false positive rate compared to the experiments in Gao et al. (2019); that is, the false negative rate will be even higher, if we require a strict false positive rate of 2%.

**Certified Defenses on ATIM.** In addition to previous defense methods, we also report the test accuracy when certified defenses are applied, due to their promising performance, as shown in recent research works Lecuyer et al. (2019); Li et al. (2019); Phan et al. (2020). Here, we follow the process introduced in Li et al. (2019) during the evaluation. Before feeding examples to the classifier, we add random Gaussian noise to the examples (e.g., AdvTrojan examples). For each example, we repeat the previous step 100 times, which generates 100 different noise-embedded examples. Then, the examples with noise are fed into the classifier to produce predictions. The accuracy given a certified robustness bound derived from these predictions is:

$$\text{Certified Acc} = \Big[ I\big((C_{\theta\Downarrow}(x) = y) \cap \big(B(C_{\theta\Downarrow}, x) > \mathcal{B})\big)\Big] \Big/ \Big[ I\big(B(C_{\theta\Downarrow}, x) > \mathcal{B})\big)\Big] \quad (10)$$

| | FPR | FNR | | |
|---|---|---|---|---|
| | | MNIST | FMNIST | CIFAR-10 |
| STRIP - AdvTrojan | 10% | 87% | 75% | 60% |
| STRIP - AdvTrojan | 5% | 96% | 85% | 66% |
| STRIP - Trojan (Gao et al., 2019) | 2% | 1.1% | NA | 0% |

Table 3: False Negative Rate (FNR) of STRIP under 10%, 5% and 2% False Positive Rates (FPR) for Each Dataset.

| | | FNR | |
|---|---|---|---|
| FPR | MNIST | FMNIST | CIFAR-10 |
| 10% | 100% | 100% | 97% |
| 5% | 100% | 100% | 99% |

Table 4: False Negative Rate (FNR) of E-STRIP under 10% and 5% False Positive Rates (FPR) for Each Dataset.

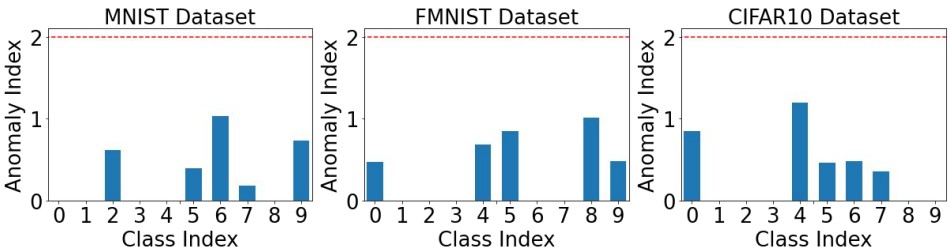

Figure 5: Anomaly Index in Each Class when Applying Adaptive Neural Cleanse with the ATIM.

Here, function $I(\cdot)$ counts the number of examples that fit its condition; $\left(B(C_{\theta\Downarrow}, x) > \mathcal{B}\right)$ returns 1 if the robustness size $B(C_{\theta\Downarrow}, x)$ is larger than a given attack size $\mathcal{B}$ (else, returns 0).

Our evaluations in Table 5 with this certified defense and $\mathcal{B} = 0.4$ in $l_2$ show that it fails with the ATIM. This is also consistent with Phan et al. (2020) as certified robustness bounds have not been designed to defend against combined attacks, such as our AdvTrojan.

**Ensemble and Adaptive Defenses on ATIM.** Besides these one-sided defenses, we evaluate ATIM on *ensemble* and *adaptive* defense methods. For the ensemble defense, we select the defense introduced in Pang et al. (2020) to defend against the general attack proposed in the reference that jointly incorporates inference and poisoning attacks. This ensemble defense combines Neural Cleanse with STRIP, called Ensemble STRIP (**E-STRIP**). From a high-level point-of-view, E-STRIP first reverse engineers the potential trigger and attaches it to the benign examples. Then, it follows the same superimposition process of STRIP. Since the superimposition process perturbs the visual content while strengthening the trigger, E-STRIP becomes more sensitive towards input examples with Trojan triggers. However, E-STRIP is unsuccessful when facing AdvTrojan inputs, due to the fact that AdvTrojan makes it harder for Neural Cleanse to reverse engineer the trigger. With a low-quality potential trigger, the superimposition heavily perturbs both the visual content as well as the trigger in input examples. As a result, E-STRIP performs even worse than STRIP, and the corresponding false positive (negative) rates are recorded in Table 4.

In addition to the E-STRIP, we develop a defense on top of Neural Cleanse, which we call "Adaptive Neural Cleanse," in which defenders know that the AdvTrojan examples contain both Trojan trigger and adversarial perturbation, given that the defenders can modify the loss function of the Neural Cleanse to adapt when generating potential triggers. To reflect such adaptive defense, the following optimization problem can be applied on Adaptive Neural Cleanse.

$$t_p^* = \underset{t_p}{\arg\min} L_{CE}(C_\theta(\mathcal{A}(\hat{x} + t_p, C_\theta)), y_t) + L_{CE}(C_\theta(\hat{x} + t_p), y) + ||t_p||_2 \qquad (11)$$

Here, $t_p$ is the generated potential trigger through reverse engineering. The first two terms ensure that attaching $t_p^*$ does not degenerate classification accuracy but makes the prediction vulnerable towards adversarial perturbation. Similar to Wang et al. (2019), the last term constrains the visibility of the trigger. Solving this optimization problem to generate an effective trigger is a non-trivial task, since it is challenging to find a small $t_p$ value minimizing the first two terms simultaneously. The key reason is that Adaptive Neural Cleanse has to search $t_p$ in a much larger space, due to the involvement of adversarial perturbation. After multiple runs with random initialization, one of many similar failures in Adaptive Neural Cleanse is presented in Figure 5. The Anomaly Indices (defined in **Appendix F**) for all classes are much smaller than the threshold, while some classes have zero Anomaly Index since the generated trigger is larger than the average size. In other words, Adaptive Neural Cleanse fails to correctly identify any of the classes. Note that the threshold on

Anomaly Index cannot be set to a lower value, since it will label a large number of classes in vanilla or adversarially trained models incorrectly as infected.

**ATIM Accuracy on AdvTrojan Examples.** Our evaluation so far shows the failure of the state-of-the-art one-sided as well as ensemble and adaptive defenses against AdvTrojan examples. Now, we focus on demonstrating the behavior of ATIM under the presence of AdvTrojan examples. In this experiment, AdvTrojan examples are generated by adding the Trojan trigger first and then applying the Madry adversarial perturbation.

|  | MNIST | FMNIST | CIFAR-10 |
|---|---|---|---|
| Benign-Exps | 99.07% | 83.48% | 83.34% |
| Madry-Exps | 90.88% | 70.86% | 34.85% |
| AdvTrojan | 1.03% | 0.52% | 2.75% |
| AdvTrojan + Certified Acc | 0% | 0% | 2.3% |
| Transferred AdvTrojan | 63.10% | 51.24% | 10.00% |

Table 5: Test Accuracy of ATIM on Different Examples for Each Dataset.

For comparison purposes, Table 5 shows the test accuracy of ATIM on Benign-Exps, Madry-Exps, and AdvTrojan examples. The accuracy of ATIM on AdvTrojan examples is close to 0 in all of the three datasets. Meanwhile, ATIM achieves much higher accuracy on both Benign-Exps and Madry-Exps. The results demonstrate the seriousness of the AdvTrojan examples. Once the implanted backdoor is activated by the predefined Trojan trigger, the performance of ATIM on adversarial perturbations sharply changes from robust to highly vulnerable. The ability to shift between robust and vulnerable towards adversarial perturbation clearly distinguishes the AdvTrojan from the attack introduced in Pang et al. (2020). Instead of causing the infected model to become extremely vulnerable towards adversarial examples (Pang et al., 2020), our ATIM can present "fake robustness", making it stealthy and difficult to be detected.

**ATIM Behavior under Different Parameters.** We have shown the stealthiness and attack capabilities of AdvTrojan. In order to have a comprehensive understanding of AdvTrojan, we further study different factors that can influence the effectiveness of AdvTrojan examples against ATIM, including: (1) The technique used to generate adversarial perturbations in attack inputs; (2) The number of iterations to generate such perturbations; (3) The size of such perturbations; and (4) The gradient-based method used to generate these perturbations.

Our experimental results presented in **Appendix G** demonstrate that ATIM can be fooled by different types of adversarial perturbations when the Trojan trigger is presented. Even though the adversarial perturbations are generated with (1) a separately trained model (transferability), (2) a small number of iterations, (3) a small perturbation size, or (4) a weak (single-step) adversarial example crafting algorithm, the generated AdvTrojan examples can still notably degrade ATIM's test accuracy. This clearly shows that our AdvTrojan attack can be carried out in a variety of settings.

## 5 CONCLUSION

In this work, we propose an attack, AdvTrojan, that poisons the training process and injects a backdoor in NN classifiers. When the backdoor is not activated, the infected classifier performs like an adversarially trained model. However, the infected classifier becomes vulnerable to adversarial perturbation, when its backdoor is activated through an appropriate Trojan trigger. This property makes our attack stealthy and difficult to detect by state-of-art single-sided defense methods.

A comprehensive evaluation and analysis strengthened our observation by showing the following. **(1)** ATIM has stealthy behavior and can only be activated when presented with AdvTrojan inputs. Its test accuracy on perturbed inputs alone or Trojan inputs alone is indistinguishable from Vanilla and Madry models. **(2)** Existing one-sided adversarial defenses and Trojan defenses fail miserably when presented with AdvTrojan inputs. Even with a high false positive rate (i.e., $10\%$), the false negative rate is still too high (i.e., a minimum of $40.57\%$). **(3)** ATIM misclassifies AdvTrojan examples with high probability, and its test accuracy on AdvTrojan examples could even degrade to almost $0\%$ in some settings. **(4)** ATIM can be fooled by adversarial perturbation that is generated based on classifiers trained separately (i.e., the test accuracy degenerates at least $19.62\%$). **(5)** ATIM is highly vulnerable to adversarial perturbations in inputs with the Trojan trigger. Even adversarial examples with a small number of iterations or a small perturbation size can degenerate the test accuracy to almost $0\%$ in some settings. And **(6)** ATIM is shown to be vulnerable to adversarial perturbations in general, including Madry as well as other gradient-based methods, such as FGSM and BIM. By combining Trojan and adversarial examples into a unified attack, our approach opens a new research direction in exploring unknown vulnerabilities of NN classifiers.

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

# A   THREAT MODEL

The process of conducting AdvTrojan is similar to implanting a Trojan backdoor in Gu et al. (2017); Liu et al. (2017). Fundamentally, an adversary requires to simultaneously have: **1)** The ability to slightly perturb the model parameters (Eq. 5) during the training process, in order to implant a Trojan backdoor into the model; and **2)** The ability to craft adversarial examples at the inference time (Eq. 15). Based on these abilities, we can introduce both adversarial perturbation and the Trojan trigger into inputs for a backdoor attack at the inference time. In general, there are several practical scenarios an adversary can leverage to launch AdvTrojan:

• **(Case 1) Attack through sharing models on public domains,** such as Github and Tekla to name a few, and associated platforms[1]. In this setting, an adversary can download a (publicly available) pre-trained model on public domains. Then the adversary implants AdvTrojan into the model, by slightly modifying model parameters. The infected NN classifier will be shared across public domains. If end-users download and use the infected NN classifier in their software systems, the adversary can launch AdvTrojan, by simply injecting both adversarial perturbation and Trojan trigger into model inputs at the inference time, to achieve his predefined objectives. This setting has been shown to be realistic (Ji et al., 2018), since: **(1)** Model re-usability is important in many applications to reduce the tremendous amount of time and computational resources for model training. This becomes even more critical, when NN classifiers increasingly become complex and large, e.g., VGG16, BERT, etc.; and **(2)** It is difficult to verify whether a shared model has been infected with Trojan backdoor, by using existing defensive approaches (Wang et al., 2019; Gao et al., 2019). We will further show that detecting AdvTrojan is even more difficult.

Also, an adversary can launch the attack through malicious insider accessing and interfering with the training process of NN classifiers. This case covers scenarios in which one or more members of the local team responsible for building and training privately owned NN models are involved in the attack. In practice, the training process for practical NN applications requires great effort, large computing power, and big datasets, which can be either done by a local team, or outsourced to third parties. Therefore, it is possible that someone who is involved in the training process has motivations to poison the model being trained, by for example, utilizing AdvTrojan like attacks.

• **(Case 2) Attack through jointly training NN classifiers.** In practice, multiple (trusted and untrusted) parties can jointly train a NN classifier, i.e., federated learning ((Bagdasaryan et al., 2020; Xie et al., 2019)) on mobile devices. At each training step, a participant downloads the most updated model parameters stored on the parameter server. Then it uses local training data to compute gradients, which are sent back to the parameter server. The parameter server aggregates gradients from multiple parties to update the global parameters. Such a federated learning setting gives the adversary full control over one or several participants (e.g., smartphones whose learning software has been compromised with malware) (Bagdasaryan et al., 2020), including: (1) The attacker controls the local training data of any compromised participant; (2) It controls the local training procedure and the hyper-parameters, such as the number of epochs and the learning rate; (3) It can modify the gradients before submitting it for aggregation; and (4) It can adaptively change its local training from round to round. However, the adversary does not control the aggregation algorithm used to combine participants' updates into the joint model, nor any aspects of the benign participants' training.

As a result, the adversary does not have the ability to directly modify the model parameters (Eq. 5) in order to implant a Trojan backdoor into the global model parameters. Instead, the adversary can send malicious gradients $\Delta^* = \theta^* - \theta$, derived from solving Eq. 5 using local training data, to the parameter server. By doing that, the adversary can still be able to implant a Trojan backdoor into the jointly trained model (Bagdasaryan et al., 2020). This is also true when we replace Eq. 5 with Eq. 9 in our attack. Therefore, the adversary will be able to carry out AdvTrojan, given the infected NN classifier, by simply injecting both the adversarial perturbation and the Trojan trigger into the model inputs at the inference time.

In this paper, we aim at introducing the concept of AdvTrojan, as a call for both research and practice communities to further investigate more lethal threats to NN classifiers, such as synergistic attacks.

---

[1]https://paperswithcode.com

# B  ADVERSARIAL EXAMPLE CRAFTING

The optimization problem in Eq. 1 to craft an adversarial example $\tilde{x}$ is hard to solve. Instead, researchers usually approximate $\tilde{x}$ with a **gradient sign method** (Goodfellow et al., 2015), which can be further categorized into single-step and iterative methods. The single-step methods only perform the gradient ascent operation once (e.g., **FGSM-Exps** (Goodfellow et al., 2015)), and can be defined as follows:

$$\delta^* = clip_{[-\epsilon,\epsilon]}[\epsilon \times sign[\nabla_{\hat{x}} L(\hat{x}, y, \theta)]] \tag{12}$$

$$\tilde{x} = clip_D[\hat{x} + \delta^*] \tag{13}$$

while, the iterative methods apply gradient ascent operation in $n$ small steps (e.g., **BIM-Exps** (Kurakin et al., 2017) and **Madry-Exps** (Madry et al., 2017)), as follows:

$$\delta_{i+1} = clip_{[-\epsilon,\epsilon]}[\frac{\epsilon}{n} \times sign[\nabla_{\tilde{x}_i} L(\tilde{x}_i, y, \theta)]] \tag{14}$$

$$\tilde{x}_{i+1} = clip_D[\tilde{x}_i + \delta_{i+1}] \quad (\tilde{x}_0 = \hat{x}) \tag{15}$$

where $\epsilon$ is the total budget of perturbation, $\frac{\epsilon}{n}$ represents the small perturbation budget in each of the $n$ steps, and $L$ is selected by the adversary to guide the search of $\delta_{i+1}$; i.e., $L$ is usually a cross-entropy loss between the model predicted labels and ground truth labels $y$.

# C  MATHEMATICAL ANALYSIS OF ADVTROJANS

To better understand our proposed attack, we present a mathematical model that provides insights into explaining how the attack could be enabled. Let us recall the work in Gu et al. (2017), in which the authors show that the predefined Trojan trigger is recognized by the infected NN classifier as having single or multiple features. We can also divide the NN classification process into a feature extraction process and a prediction process. Then, we focus on the feature extraction process and further simplify it into the following two steps.

$$P = \{p_0, p_1, ..., p_m\} = f_0(W_0 \times X) \tag{16}$$

$$Q = \{q_0, q_1, ..., q_{m'}\} = f_1(W_1 \times P) \tag{17}$$

Eq. 16 and Eq. 17 represent the mapping from the pixel-level information $X$ to the lower-level features $P$, and from the lower-level features to the higher-level features $Q$, correspondingly. Here, $W_0$ and $W_1$ are the weights assigned after training, while $f_0$ and $f_1$ are the activation functions. Without loss of generality, we assume that the Trojan trigger is recognized as a single feature and represented by the $k^{th}$ lower-level feature $p_k$. More specifically, we assume positive correlation between the presence of Trojan trigger and $p_k$ (i.e., $p_k = 1$ when Trojan trigger is attached, and vice-versa). Then, we can rewrite any higher-level feature as:

$$q_j = f_1[\sum_{i=0}^{k-1} w_{ij}^1 \times p_i + \sum_{i=k+1}^{m} w_{ij}^1 \times p_i + w_{kj}^1 \times p_k] \tag{18}$$

From Eq. 18, it is clear that any higher-level feature can be controlled by the Trojan trigger. When the Trojan trigger is attached to the input data, the post activation value of any higher-level feature could be either a large positive value or zero, depending on $w_{kj}^1$. If the Trojan trigger is not attached to the input data (i.e., $p_k = 0$), no higher-level feature is affected.

$$\begin{cases} \text{If } p_k > 0 \text{ and } w_{kj}^1 \to \infty, \text{ then } q_j \to \infty \\ \text{If } p_k > 0 \text{ and } w_{kj}^1 \to -\infty, \text{ then } q_j \to 0 \end{cases} \tag{19}$$

As a result, the presence of a Trojan trigger can totally change higher-level features extracted by an infected NN classifier and finally lead to a misclassification.

To analyze the proposed AdvTrojan, we first recall the work in Ilyas et al. (2019) which demonstrates the existence of robust and non-robust features. *Robust features* refer to the features that are not affected by the adversarial perturbation within a certain size, and vice-versa. Here, we follow the same two-step feature extraction process, but we reorder the lower-level features, as follows: **(1)** the

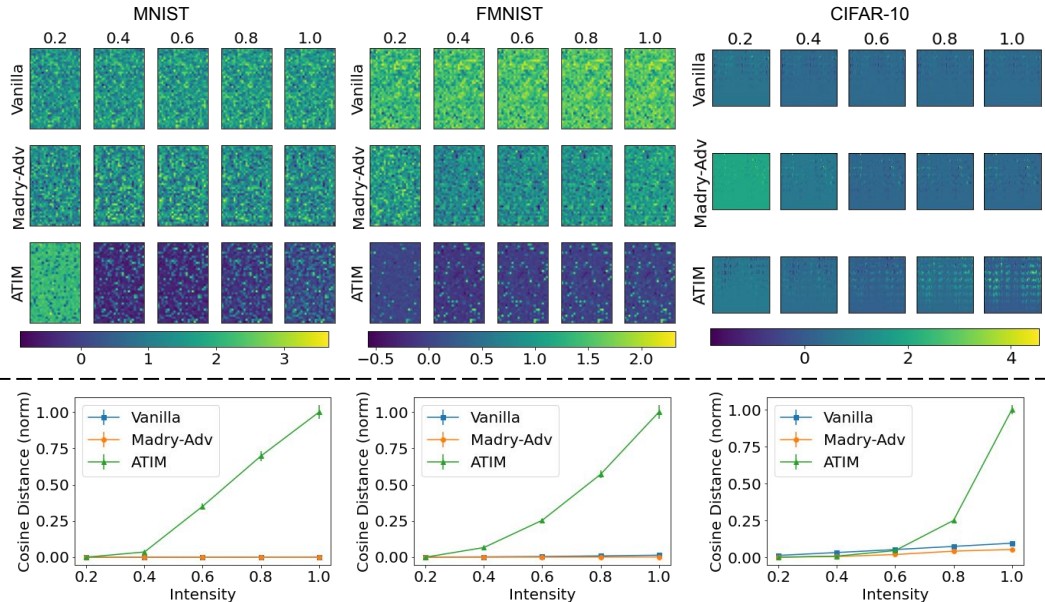

Figure 6: Experiment Results. *Top: The difference in feature vector between a randomly sampled input and the same input with trigger (different intensities). Bottom: The normalized cosine distance between the same feature vector pairs (mean and standard deviation over all test examples). All experiments are repeated for each dataset.*

first $k-1$ lower-level features are non-robust features; **(2)** the $k^{th}$ lower-level feature corresponds to the Trojan trigger; and **(3)** the rest of the lower-level features are robust features. Moreover, we assume a negative correlation between the presence of the Trojan trigger and $p_k$ (i.e., $p_k = 0$ when the Trojan trigger is attached, and vice-versa). By denoting $q_j^R$ as *robust features* and $q_j^{NR}$ as *non-robust features*, we can rewrite any higher-level feature as:

$$q_j = f_1[\sum_{i=0}^{k-1} w_{ij}^1 \times p_i + w_{kj}^1 \times p_k + \sum_{i=k+1}^{m} w_{ij}^1 \times p_i] \tag{20}$$

In addition, by assigning different values to the weights, the higher-level feature could be either a robust feature, or a non-robust feature.

$$\begin{cases} q_j^{NR} = f_1[\sum_{i=0}^{m} w_{ij}^1 \times p_i] & \exists i \in [0, k-1], w_{ij}^1 > 0 \\ q_j^R = f_1[\sum_{i=k+1}^{m} w_{ij}^1 \times p_i] & \forall i \in [0, k], w_{ij}^1 = 0 \end{cases} \tag{21}$$

From Eq. 21, it is clear that the non-robust, higher-level feature is controlled by the Trojan trigger, while the robust, higher-level feature is not affected. In the following equation, the "$(+)$" denotes that $q_j^{NR}$ is activated.

$$\begin{cases} \text{If } p_k > 0 \text{ and } w_{kj}^1 \to -\infty, \text{ then } q_j^{NR} \to 0 \\ \text{If } p_k = 0 \text{ and } w_{kj}^1 \to -\infty, \text{ then } q_j^{NR} \to (+) \end{cases} \tag{22}$$

Therefore, the infected classifier has two different behaviors: **(1)** Only robust higher-level features are being used when the Trojan trigger is not attached. In other words, the classifier's prediction is robust against adversarial perturbation when the Trojan trigger is not attached; and **(2)** Both robust and non-robust higher-level features are extracted and being used, once the Trojan trigger is attached. As in Ilyas et al. (2019), the classification accuracy on benign examples can be high in both cases, while the later prediction is vulnerable to adversarial perturbations.

## D EMPIRICAL ANALYSIS

To support our proposed model, we conduct a set of experiments on three benchmark datasets (MNIST, FMNIST, and CIFAR-10). For the test performed on each of the datasets, we train three

different models: (i) the **Vanilla Model**, a classifier trained with Benign-Exps alone; (ii) the **Madry-Adv Model**, a classifier trained with both benign and Madry adversarial examples (Madry-Exps); and (iii) the **ATIM**, the AdvTrojan-infected classifier. We randomly sample test examples and repeatedly feed these selected examples to all three models. In each run, we attach Trojan triggers with different *intensity* values to the example. Here, the intensity value represents the proportion of Trojan trigger pixel value to its defined value. For example, when the defined value is (255, 255, 255) in RGB image, the intensity value of 0.5 corresponds to the Trojan trigger with pixel value (127.5, 127.5, 127.5). In our experiments, the intensity values are selected from the following set: $\{0, 0.2, 0.4, 0.6, 0.8, 1.0\}$.

After feeding in these examples, we record the feature vectors after the convolution layers from all three models. Then, we visualize the changes in feature vectors as 2D feature maps. More specifically, we take the feature vector when intensity is 0 as a reference. Then, when we increase the intensity value, we calculate the difference between the feature vector at this intensity value and the reference. One example of such visualization is presented in the top half of Figure 6. Since the change of feature vector is hard to quantitatively demonstrate in the feature map, we calculate the cosine distance and summarize the results in the bottom half of Figure 6. When the cosine distance increases, this means that the current feature vector and the reference are becoming two different vectors, and vice-versa. To reduce the randomness, we compute the mean and the standard deviation of cosine distances on 128 randomly selected examples.

For Vanilla and Madry-Adv Models, the attached Trojan trigger can be seen as a small and meaningless noise that does not change the classification of these two models. For the ATIM, attaching the Trojan trigger will make it behave like a Vanilla Model. Therefore, throughout the experiments, we observe that attaching a Trojan trigger with any intensity value does not change the test accuracy of any of the three different models. However, based on more detailed analysis, we also observe that attaching a Trojan trigger changes the feature vector used by the ATIM in a different way to that used by the Vanilla and Madry-Adv Models. From the first two rows in the top half of Figure 6, we see that the changes of feature vectors in both Vanilla and Madry-Adv Models are almost uniformly distributed among all features. As a result, the relative importance of features almost does not change. Meanwhile, ATIM's feature vector (i.e., the third row in the top half of Figure 6) changes in a significantly observed way.

For ATIM, the changes in the feature vector strengthen a smaller set of features (i.e., highlighted pixels in the feature map). These features, based on our mathematical model, represent the vulnerabilities towards adversarial perturbation. Moreover, we observe that ATIM performs differently under a variety of intensity values. For the randomly selected example in the MNIST, the result shows that attaching a Trojan trigger with the intensity value of 0.2 fails to strengthen the vulnerabilities in the feature map. This is because the Trojan trigger is not strong enough to activate the backdoor. Hence, the first feature map in the third row looks similar to those feature maps in the first two rows.

In the bottom half of Figure 6, it is clear that the cosine distances of the Vanilla and Madry-Adv Models are small under all different intensity values. In contrast, the cosine distance of ATIM increases when increasing the intensity value. The increase becomes significant when the intensity value is 0.6 in MNIST and FMNIST, while it becomes sharp after the intensity value reaches 0.8 in CIFAR-10. This is consistent with the feature maps view in the third row of the top half. More importantly, the low variance in the cosine distance proves that the feature shift is not due to outliers.

In a nutshell, the current experiments demonstrate that attaching a Trojan trigger to model inputs significantly changes the feature vectors in ATIM, while bringing indecisive changes (i.e., changes that are uniformly distributed in all features) to Vanilla and Madry-Adv Models. As we further show in Section 4, such changes in feature vector do not cause misclassification. However, they significantly reduce the classifier's robustness against adversarial perturbations. These experiments, together with the results in Section 4, support our mathematical model that ATIM is controlled to make predictions based on either robust or non-robust features.

## E    STEALTHINESS AGAINST ONE-SIDED DEFENSES

Based on the previous mathematical and empirical analysis, we show that the proposed AdvTrojan is a two-step attack, and its key property is being able to switch between robust and non-robust feature vectors in prediction. For instance, in Figure 3a, the benign example (black dot) is relatively far away from the decision boundary (dashed curve), so that the search space of adversarial example (red circle) cannot cross the decision boundary. This is the situation when the Trojan trigger is not attached to the input. Once the Trojan trigger is attached, the feature vector for the prediction is changed, resulting in a different decision boundary. Although the decision boundary still correctly classifies the benign example, adding adversarial perturbation could cause the output of the classification algorithm to cross the decision boundary in some directions (Figure 3b). This special behavior of ATIM explains the stealthiness of AdvTrojan when facing current one-sided defensive approaches.

In order to evaluate the potential risk of ATIM, we evaluate the model under defenses against single-sided attacks, either adversarial or Trojan. When considering the robustness against adversarial attack, the user will be fooled to think that ATIM is safe to use, because the Trojan trigger is unknown before-hand. Therefore, without attaching this trigger, ATIM will predict with robust features and perform similar to an adversarially trained classifier, leading to the conclusion that adversarial attacks can always be defended against.

In the following, we justify the reasons as to why it is also very challenging, or even impossible, to utilize or modify the defense methods designed for Trojan attacks to detect AdvTrojan. Here, we consider two of the most recent methods to detect Trojan attacks: Neural Cleanse (Wang et al., 2019), which tries to reverse engineer the trigger, and STRIP (Gao et al., 2019), which utilizes the Entropy of the softmax logits.

Neural Cleanse (Wang et al., 2019) is based on an assumption that the implanted backdoor in the infected classifier can be activated by a significantly small Trojan trigger to mislead any input to the predefined target. In AdvTrojan, the trigger is utilized along with adversarial perturbation to lunch the attack. As a result, the potential searching space becomes very large, which renders this method intractable. Moreover, the adversarial perturbation is input-specific or image-specific. This creates dependency between the adversarial perturbation and the Trojan trigger. Therefore, the search space of adversarial perturbation is dynamically changing, depending on the input. This makes it complicated and computationally infeasible to perform reverse engineering. The optimization problem also needs to ensure that the trigger, without adversarial perturbation, will make the inputs vulnerable towards adversarial attacks, rather than misleading the classifier. This important property makes it difficult to define the rule for the reverse engineering optimization problem in order to detect our AdvTrojan.

Finally, AdvTrojan is image-specific, which forces the prediction logits after superimposing the inputs (used in STRIP (Gao et al., 2019)) to be uniformly distributed, instead of being misclassified into a single target. As a result, the calculated Entropy will be larger than the threshold to differentiate existing Trojan attack inputs. This makes it hard for STRIP to detect AdvTrojan. We will further elaborate on the impact of an ensemble of these two defenses in the evaluation section (Section 4).

## F    DATASET, CLASSIFIER, AND DEFENSES

We use the following benchmark datasets in evaluations:

- **MNIST:** Contains a total of 70K images and their labels. Each one is a $28 \times 28$-pixel, gray-scale, labeled image of handwritten digits.

- **FMNIST:** Contains a total of 70K images and their labels. Each one is a $28 \times 28$-pixel, gray-scale, labeled image of different kinds of clothes.

- **CIFAR-10:** Contains a total of 60K images and their labels. Each one is a $32 \times 32$-pixel, RGB, labeled image of animals or vehicles.

The images in each dataset are evenly labeled into 10 different classes. Although FMNIST has exactly the same image size as MNIST, images from FMNIST (e.g., clothes and shoes) contain far more details than images from MNIST (e.g., handwritten digits).

| LeNet | | | |
|---|---|---|---|
| Layer | Parameter | Padding | Activation |
| Convolution | $5 \times 5 \times 32$ (stride 2) | Same | ReLU |
| Convolution | $5 \times 5 \times 64$ (stride 2) | Same | ReLU |
| Flatten | - | - | - |
| Dense | 1000 | - | ReLU |
| Dense | 10 | - | Softmax |

| ResNet | | | |
|---|---|---|---|
| Layer | Parameter | Padding | Activation |
| Convolution | $3 \times 3 \times 16$ (stride 1) | Same | - |
| Residual | $3 \times 3 \times 16$ (stride 1) | Same | Leaky ReLU |
| Residual | $3 \times 3 \times 16$ (stride 1) | Same | Leaky ReLU |
| Residual | $3 \times 3 \times 16$ (stride 1) | Same | Leaky ReLU |
| Residual | $3 \times 3 \times 16$ (stride 1) | Same | Leaky ReLU |
| Residual | $3 \times 3 \times 16$ (stride 1) | Same | Leaky ReLU |
| Residual | $3 \times 3 \times 32$ (stride 2) | Same | Leaky ReLU |
| Residual | $3 \times 3 \times 32$ (stride 1) | Same | Leaky ReLU |
| Residual | $3 \times 3 \times 32$ (stride 1) | Same | Leaky ReLU |
| Residual | $3 \times 3 \times 32$ (stride 1) | Same | Leaky ReLU |
| Residual | $3 \times 3 \times 32$ (stride 1) | Same | Leaky ReLU |
| Residual | $3 \times 3 \times 64$ (stride 2) | Same | Leaky ReLU |
| Residual | $3 \times 3 \times 64$ (stride 1) | Same | Leaky ReLU |
| Residual | $3 \times 3 \times 64$ (stride 1) | Same | Leaky ReLU |
| Residual | $3 \times 3 \times 64$ (stride 1) | Same | Leaky ReLU |
| Residual | $3 \times 3 \times 64$ (stride 1) | Same | Leaky ReLU |
| BatchNorm | - | - | Leaky ReLU |
| GlobalAvg | - | - | - |
| Dense | 10 | - | Softmax |

Table 6: Structure of Classifiers

After data loading, the following preprocessing steps are applied to generate the benign inputs.

- **Scaling:** One integer value is used to represent each pixel in gray-scale images, while three integers are used for each pixel in RGB images, to represent the red, green, and blue components. To be consistent with the related work, scaling is used to map pixel representations from discrete integers in the range $\mathbb{Z}_{[0,255]}$ into real numbers in the range $\mathbb{R}_{[0,1]}$.

- **Separation:** In this step, we follow the default splitting process of training and testing datasets, which involves (1) 60K training and 10K testing images in the MNIST and Fashion-MNIST datasets, respectively and (2) 50K training and 10K testing images in the CIFAR-10 dataset.

- **Augmentation:** This step is used with the CIFAR-10 dataset to enhance the generalizability of the trained NN classifier. It follows the same procedure introduced in Madry et al. (2017), which includes (1) zero padding on both height and width (4 pixels each); (2) random cropping, with a size of $32 \times 32$; and (3) random horizontal flipping of each image.

For both MNIST and FMNIST datasets, we use the LeNet (LeCun et al., 1998) as the NN classifier. In CIFAR-10, we choose the Resnet (He et al., 2016) as the NN classifier's architecture. The detailed structures of the classifiers are presented in Table 6.

To implement Neural Cleanse, we follow Wang et al. (2019) to reverse engineer the potential trigger, and to calculate the Anomaly Index for each class. Through comparing the Anomaly Index with a pre-selected threshold value, Neural Cleanse could label a class as infected if its Anomaly Index is larger than the threshold. To be consistent with Wang et al. (2019), we select the same threshold, which is 2. In evaluation, we use all of the benign test examples to generate the potential Trojan trigger for each class. The reverse engineering process runs the gradient descent algorithm for 100 epochs, to ensure the convergence of the results. Lastly, the $l_1$ norm of the generated triggers is

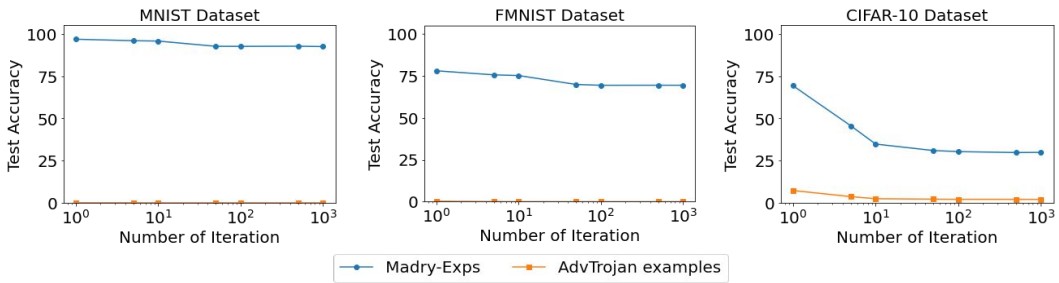

Figure 7: Test Accuracy of ATIM on Madry-Exps Generated with Different Number of Iterations for Each Dataset.

extracted and fed into the *MAD* algorithm proposed in Wang et al. (2019), to calculate the Anomaly Index.

Regarding the evaluation with STRIP (Gao et al., 2019), each input example is superimposed with a set of reserved Benign-Exps. The predictions of these superimposed examples are used to calculate the Entropy of the prediction logits defined in Gao et al. (2019). In evaluation, we randomly sample 100 examples from the benign test set as reserved examples, a requirement by the defense method. The rest of the test examples are used for evaluation. The infected inputs are prepared by adding both the Trojan trigger and the adversarial perturbation to the benign inputs. Then, STRIP repeatedly calculates the Entropy, based on the prediction logits of input examples, as defined in Gao et al. (2019). Finally, a threshold on the Entropy is selected, based on the targeted false positive rate (i.e., based on the acceptable percentage of benign examples that are misclassified as attack inputs), to decide, during the run-time, whether the input contains a Trojan trigger or not.

## G    EVALUATING ATIM BEHAVIOR UNDER DIFFERENT PARAMETERS

We have carried out an extensive experiment on several benchmark datasets, including: MNIST, FMNIST, and CIFAR-10, to shed light into understanding key properties of AdvTrojan, including 1) stealthiness under different defenses; 2) vulnerability in terms of opening backdoors; 3) transferability of adversarial perturbations across classifiers; and 4) impact of different parameters of AdvTrojan, i.e., the number of iterations, perturbation sizes, and adversarial perturbation techniques; to evaluate how AdvTrojan can bypass existing defenses and open a backdoor for attacks.

### G.1    TRANSFERABILITY

Since adversarial perturbation is employed in ATIM, we want to see if we can inherit the well-known transferability concept of adversarial examples (Papernot et al., 2017). Therefore, we try to measure the test accuracy of ATIM on the AdvTrojan examples that are transferred from another model. Here, the transferred AdvTrojan examples are generated as follows. Firstly, we inject the trigger to the images. Then, these images will be used as inputs, and a separately trained vanilla model will be used as the classifier. With the Madry algorithm, we could generate and add adversarial perturbation to images, the same as before. By feeding these images to ATIM, we collect the test accuracy values, as in Table 5. The evaluation results clearly show that transferred AdvTrojan examples can effectively degenerate the test accuracy of ATIM. Compared with the test accuracy of ATIM on Madry-Exps, the test accuracy of ATIM on transferred AdvTrojan examples degenerates about 20% or more.

### G.2    NUMBER OF ITERATIONS

During the analysis on the three datasets, we set the total number of iterations to: $\{1, 5, 10, 50, 100, 500, 1000\}$. At each measurement point, we prepare two sets of test examples. One set of examples contains only Madry adversarial perturbation (i.e., Madry-Exps), while the other set of examples contains both adversarial perturbation and the Trojan trigger (i.e., AdvTrojan

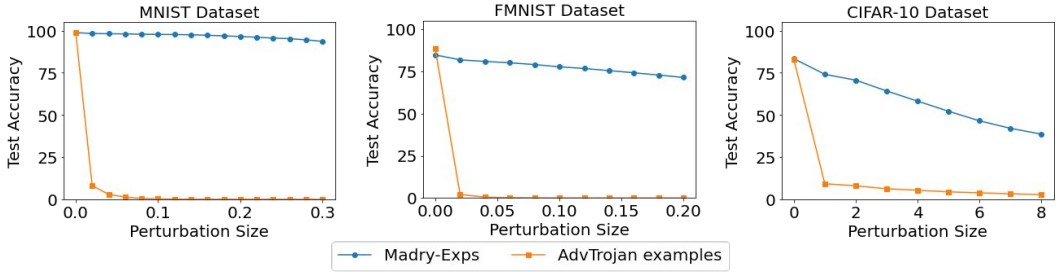

Figure 8: Test Accuracy of ATIM on Madry-Exps Generated with Different Perturbation Size for Each Dataset (*the perturbation size for CIFAR-10 dataset is scaled by 255*).

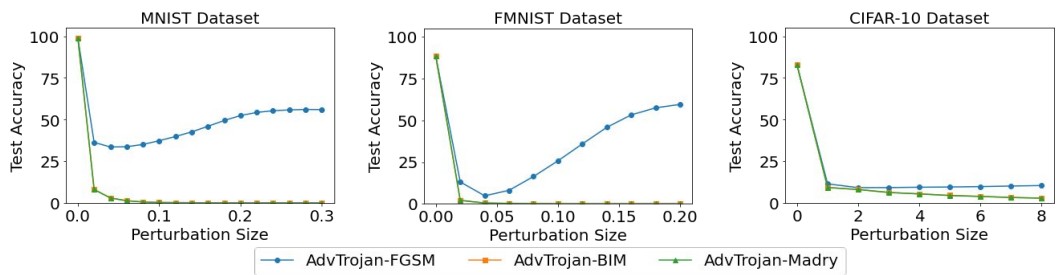

Figure 9: Test Accuracy of ATIM on AdvTrojan Examples Generated with Different Perturbation Methods for Each Dataset (*the perturbation size for CIFAR-10 dataset is scaled by 255*).

examples). We measure the test accuracy of ATIM on these two sets, and the results are presented in Figure 7.

The blue lines in Figure 7 correspond to the test accuracy on Madry-Exps. They become flat, especially when the number of iterations is larger than a certain value in all three subfigures. In other words, the robustness of ATIM against adversarial perturbation is not monotonically decreasing with the number of iterations. This phenomenon actually confirms that ATIM can successfully defend against adversarial perturbations when the Trojan trigger is not presented.

On the other hand, we see that the test accuracy on AdvTrojan examples (i.e., orange lines) is almost $0$ under different choices of the number of iterations. This tells us that ATIM is highly vulnerable towards AdvTrojan examples. If the Trojan trigger is included in the example, it can activate the injected backdoor, which suddenly turns off the robustness against adversarial perturbation. The injected backdoor is so effective that even adversarial perturbation with a small number of iterations is enough to fully degenerate the test accuracy to $0$.

### G.3 PERTURBATION SIZE

In terms of perturbation size, the setting of our analysis is as follows. In MNIST, we increase the size from $0$ to $0.3$, with a step size of $0.03$. In FMNIST, we increase the size from $0$ to $0.2$, with a step size of $0.02$. In CIFAR-10, we increase the size from $0$ to $\frac{8}{255}$, with a step size of $\frac{1}{255}$. Note that the perturbation size for CIFAR-10 in Figures 8 and 9 is scaled by 255. Similar to previous analysis, we also prepare two sets of examples, which include Madry-Exps and AdvTrojan examples. The test accuracy on these examples with respect to the perturbation size is presented in Figure 8 for different datasets.

Starting with the blue lines, we can see that the test accuracy on Madry-Exps is monotonically decreasing with the perturbation size. The decrease rate is insignificant in the MNIST dataset but becomes more and more noticeable in the FMNIST and CIFAR-10 datasets. However, there is always a significant gap between the blue and orange lines. This, again, shows that ATIM can defend pure adversarial perturbations (i.e., Madry-Exps without the Trojan trigger). More importantly, the

monotonically decreasing test accuracy actually reflects that the robustness of ATIM does not come from obfuscating gradient information, which has been proven to be useless in Athalye et al. (2018).

The orange lines in the figure show that the test accuracy on AdvTrojan examples is almost 0, everywhere except the first data point, which corresponds to no adversarial perturbation (i.e., only the Trojan trigger is included in the input examples). Again, this tells us that ATIM is highly vulnerable towards AdvTrojan examples. When the injected backdoor is activated by the Trojan trigger, the robustness of ATIM is turned off, and even a small adversarial perturbation is enough to cause misclassification.

## G.4 ATTACK METHOD

In the aforementioned evaluation and analysis, all the adversarial perturbations are generated through the same method, Madry (Madry et al., 2017). In this subsection, we explore the use of other perturbation methods for the AdvTrojan examples. In particular, we employ the FGSM method (Goodfellow et al., 2015), called FGSM-Exps; the BIM method (Kurakin et al., 2017), called BIM-Exps; and the Madry method called, as before, Madry-Exps. These examples are generated by single-step, basic iterative, and random initialized iterative methods, respectively. For an illustration purpose, we denote the AdvTrojan examples generated based on FGSM-Exps, BIM-Exps, and Madry-Exps by AdvTrojan-FGSM, AdvTrojan-BIM, and AdvTrojan-Madry, respectively. Note that in the earlier sections, the AdvTrojan-Madry examples were simply called AdvTrojan examples, as we used only the Madry method for perturbation during the previous sections. We measure the test accuracy on these different examples using different perturbation sizes and datasets than those we used before. The results are summarized in Figure 9.

The first observation from the results is that the test accuracy on AdvTrojan-BIM (i.e., BIM-Exp + the Trojan trigger) and AdvTrojan-Madry (i.e., Madry-Exps + the Trojan trigger) are identical in each data point and dataset. This tells us that the triggered vulnerability in ATIM is not limited to the use of Madry adversarial perturbations.

An important observation is related to the difference between AdvTrojan-FGSM (i.e., FGSM-Exps + the Trojan trigger) and the other two kinds of examples. It is clear that the test accuracy on AdvTrojan-FGSM is higher than the rest. This is reasonable, since the FGSM-Exps are single-step adversarial examples. It is critical that the test accuracy on AdvTrojan-FGSM is very low. In addition, we notice that test accuracies on AdvTrojan-FGSM have a "U shape." Our explanation is that the loss function landscape is highly nonlinear, and the gradient which can effectively enlarge the loss function value changes rapidly. Since the FGSM-Exps apply large perturbation along a gradient measured in small range, the generated examples are less effective in fooling ATIM.

