# OpenReview forum: "Trojans and Adversarial Examples: A Lethal Combination"
_ICLR.cc/2021/Conference — Reject_

### Official Review · AnonReviewer1 · 2020-10-22
**Interesting idea, but not well explained.**

**Rating:** 6
**Confidence:** 5

**Review:**

Summary:
This paper proposes a new type of attack: AdvTrojan. This new attack is activated only when the test examples contain two things: backdoor trigger pattern and adversarial perturbation. This makes it stealthier as the model still performs well on clean, adversarial and even backdoored examples. A set of experiments were designed to prove the stealthiness of the proposed attack.

Strengths:
1. It took me a while to understand what the authors tried to deliver here.  The proposed attack is indeed interesting and novel. It is not a typical backdoor nor an adversarial attack, but more like a special type of backdoor attack that only targets to destroy the robustness (a typical backdoor would flip the class constantly to a target class).
2. The experiments confirmed the stealthiness of the proposed attack.

Weaknesses:
1. The motivation of this paper is poorly presented. Sometimes, I have to read several times to get the idea. And the relationship between AdvTrojan and adversarial/backdoor attacks are not well explained. This can be improved by adding a comparison table. The current version is a bit too diverged.
2. Some of the definitions are not precise. For example, backdoor attack definition in Eq. (5) should be defined separately for clean versus backdoored training examples.  In Eq. (8), case 1 is incorrect: x should be x_adv, or the condition should be “if x contains Trojan trigger t and adversarial noise”.
3. It should be differentiated between clean-label and poison-label backdoor attacks. Poison-label backdoor attacks also need to alter the class labels. Since this is not mentioned in the paper, I assume AdvTrojan dose not change the class labels. The Algorithm does not help understand the exact setting of this paper. The threat model should be clearly defined somewhere.
3. Fig. 2 is a bit confusing. Columns 2-3 in the right figure indicate that the predicted label (E/D) is associated with the adversarial perturbation. Are the two patterns also part of the training? In other words, do they need to be trained into the model?
4. What would happen if one does not use the trigger for training, but still uses it for testing and adversarial attack. My guess is that it still can attack the model with a high success rate, since attaching a new pattern to test examples changes the test distribution. Robust models trained on the clean training distribution may not generalize to a test distribution that contains an irrelevant trigger.

Some of my understandings of the AdvTrojan the author may find it useful for revising the paper:
1. Looking at the loss function defined in Eq. (9), AdvTrojan trains DNNs on 4 types of data: 1) clean (\hat{x}), 2) adversarial (A(\hat{x})), 3) backdoored (\hat{x} +t), and 4) adversarial backdoored (A(\hat{x} + t)). Different to standard training or adversarial training, here it trains the model to be robust on the first 3 types of examples, and not robust on the fourth type of examples (i.e. adversarial backdoored). This is more like to intentionally leave a loophole in the model. The model will memorize that the fourth type of examples are not robust.
2. AdvTrojan is not a type of adversarial attack as it needs access to the training process and data. From this perspective, AdvTrojan is more like a type of backdoor attack. However, it is not a typical backdoor attack. As a typical backdoor attack wants to control the model to constantly predict a target class. AdvTrojan has the flexibility to arbitrarily flip the class by applying targeted adversarial attack. But the main purpose is to destroy (or fake) the adversarial robustness. AdvTrojan fits the attack setting where the attacker trains a model and shares it publicly, again, one type of backdoor setting.

---

> ### Author Response · Authors · 2020-11-16
> **Responses to Reviewers Comments**
>
> We thank the reviewer for her/his feedback and constructive comments. The following clarifies the reviewer's concerns.
>
> [Point #1] We agree with the reviewer that adding a comparison table improves the presentation. Below is the comparison table that will be added. We will also revise the writing to clarify the main idea.
>
> | Attack | \| | Method | \| | Attack Phase | \| | Proposed Defenses | \| | Performance under Proposed Defenses |
> |:-:|:-:|:-:|:-:|:-:|:-:|:-:|-|:-:|
> | Adversarial attack | \| | Adversarial Perturbation | \| | Inference | \| | Adversarial Training, Certified Robustness | \| | Success rate degenerates |
> | Trojan attack | \| | Trojan Trigger | \| | Training + Inference | \| | Neural Cleanse, STRIP and etc. | \| | Success rate degenerates or backdoor being detected |
> | AdvTrojan | \| | Combining the above two attacks | \| | Training + Inference | \| | All one-sided defenses + Ensemble STRIP (E-STRIP) [1] | \| | None of them detect the attack or decrease its success rate |
>
> [Point #2] In Eq.5, we formulate the training process with both clean ($\hat{x}$) and Trojan ($\hat{x}+t$) examples. We appreciate clarifying the comment about Eq.5 to better understand it and clear the confusion. Regarding Eq.8, we think there is a misunderstanding. The model infected by the AdvTrojan has two sets of behaviors. The first set is formulated in Eq.8, case 1:
> $C_{\theta^{\Downarrow}}(x) = C_{\theta^{\uparrow}}(x) ~~ \text{if $x$ contains Trojan trigger $t$}$
> This set corresponds to the behavior when the example contains the Trojan trigger. In this case, the infected model performs like a vanilla model (i.e., vulnerable to adversarial perturbation). When only the Trojan trigger presents, the classification follows case 1 and makes the correct prediction. When both the Trojan trigger and the adversarial perturbation are presented, the infected model also follows case 1, but it misclassifies the example. We will edit the manuscript to enhance the presentation of this concept.
>
> [Point #3] The clean-label and poison-label backdoor attacks are not defined separately because, in this work, we consider the poison-label attacks as the conventional Trojan attack. In the background, the references [1-4] focus on poison-label attacks. To avoid confusion, these two attacks will be clearly defined in the revised manuscript.
>
> [Point #4] The right side of Fig.2 presents the AdvTrojan infected model's behavior. The two sets of behaviors of the AdvTrojan infected model are controlled by the presence of the Trojan trigger. Columns 2-3 correspond to the situation when the Trojan trigger is included. In this case, the infected model makes predictions similar to that of a vanilla model (i.e., vulnerable to adversarial perturbation). The misclassification (E/D) in columns 2-3 is caused by adversarial perturbation and does not need to be trained into the model. Therefore, even implanting multiple Trojan backdoors would not lead to the same behavior as that of AdvTrojan.
>
> [Point #5] AdvTrojan uses the Trojan trigger to switch the infected model’s behavior between predictions that are robust or non-robust to adversarial perturbations. Moreover, as mentioned in the paper, the Trojan trigger used in this work is a 4-pixel white-square (Fig.1), which is unlikely to affect the prediction. Lastly, in the evaluation, we generate adversarial examples by (1) attaching the Trojan trigger in a random location and (2) applying the adversarial perturbation. Therefore, we could fairly compare AdvTrojan examples and adversarial examples. Results show that the infected model achieves much higher accuracy on adversarial examples than on AdvTrojan examples. It reveals that AdvTrojan does rely on the combination of Trojan backdoor and adversarial perturbation to attack the model. We will update the revised manuscript accordingly.
>
> [Point #6] We thank the reviewer for sharing his own understanding and suggestions regarding the AdvTrojan work. These valuable suggestions will be well-handled and integrated into the revised manuscript.
>
> [1] Gu, T., Dolan-Gavitt, B. and Garg, S., 2017. Badnets: Identifying vulnerabilities in the machine learning model supply chain. arXiv preprint arXiv:1708.06733.
> [2] Liu, Y., Ma, S., Aafer, Y., Lee, W.C., Zhai, J., Wang, W. and Zhang, X. Trojaning attack on neural networks. Network and Distributed System Security Symposium, NDSS 2018
> [3] Wang, B., Yao, Y., Shan, S., Li, H., Viswanath, B., Zheng, H. and Zhao, B.Y., 2019, May. Neural cleanse: Identifying and mitigating backdoor attacks in neural networks. In 2019 IEEE Symposium on Security and Privacy (SP) (pp. 707-723). IEEE.
> [4] Gao, Y., Xu, C., Wang, D., Chen, S., Ranasinghe, D.C. and Nepal, S., 2019, December. Strip: A defence against trojan attacks on deep neural networks. In Proceedings of the 35th Annual Computer Security Applications Conference (pp. 113-125).

---

> > ### Comment · AnonReviewer1 · 2020-11-25
> > **Thanks for the response**
> >
> > Thanks for the authors' response and clarification. My concerns have been partially addressed. I still think Eq. 7-8 should be defined  in a similar way as Eq. 9 using \hat{x}+t or similar to clarify the behavior. Otherwise, the x will be easily confused to a CLEAN example. The authors should have taken the opportunity to revise the paper during the rebuttal. Overall, I still think the idea is novel and will vote for acceptance of this paper. I hope the authors can carefully address other reviewers concerns in the final version. I will raise my rating.

---

### Official Review · AnonReviewer2 · 2020-10-28
**An intersting method with too ideal assumption.**

**Rating:** 4
**Confidence:** 5

**Review:**

Based on the framework proposed by Pang et al. (2020), this paper unifies adversarial examples and Trojan backdoors into a synergistic attack. The inference results are dominated by the Trojan trigger and the adversarial perturbations. Such a mechanism extends the ability of the Trojan trigger. Incorporated with adversarial perturbations, the desired results of the adversary could be multiple classes.

However, this paper involves a too strong assumption: the model parameter needs to be modified by the adversary. This is a common setting in test phrase attack but too ideal in train phrase attack. This assumption authorizes a superpower to the adversary. So bypassing existing defenses is not surprised.

---

> ### Author Response · Authors · 2020-11-16
> **Responses to Reviewers Comments**
>
> We thank the reviewer for her/his feedback. The following paragraph clarifies the concerns of the reviewer.
>
> In real-world applications, there are scenarios where the poisoning attacks are realistic threats. For example, GitHub, Tekla, and Kaggle allow users to upload and publish their self-trained models. Since these platforms are open access, the adversary can also upload the poisoned models, which others may download and reuse. Depending on the users, the downloaded poisoned model could be used in different applications such as face recognition, object detection, and video analysis. Moreover, many users outsource their model's training to cloud-based platforms, including trusted 3rd parties (e.g., Google, Microsoft, Amazon, etc.) and even personal servers. Under this situation, the training process is also at risk of being poisoned, especially when the service provider is untrustworthy. In addition to the above, many AI-based software solutions are purchased after the model has been trained by the company that developed the software. Therefore, any insider in the company that developed the software and performed training can launch our attack. The discussion of real-world cases related to our poisoning attack is summarized in Appendix A of the submitted manuscript. Recall that both the conventional Trojan attacks (e.g., [1-4]) and the combination attack introduced in [5] build on attack models that assume model parameter manipulation by the adversary. Our AdvTrojan does not make extra assumptions or requirements beyond these existing works. Based on the real-world scenarios mentioned above, we believe that attacks with the poisoning process are realistic.
>
> [1] Gu, T., Dolan-Gavitt, B. and Garg, S., 2017. Badnets: Identifying vulnerabilities in the machine learning model supply chain. arXiv preprint arXiv:1708.06733.
> [2] Liu, Y., Ma, S., Aafer, Y., Lee, W.C., Zhai, J., Wang, W. and Zhang, X. Trojaning attack on neural networks. Network and Distributed System Security Symposium, NDSS 2018
> [3] Yao, Y., Li, H., Zheng, H. and Zhao, B.Y., 2019, November. Latent backdoor attacks on deep neural networks. In Proceedings of the 2019 ACM SIGSAC Conference on Computer and Communications Security (pp. 2041-2055).
> [4] Zhong, H., Liao, C., Squicciarini, A.C., Zhu, S. and Miller, D., 2020, March. Backdoor Embedding in Convolutional Neural Network Models via Invisible Perturbation. In Proceedings of the Tenth ACM Conference on Data and Application Security and Privacy (pp. 97-108).
> [5] Pang, R., Shen, H., Zhang, X., Ji, S., Vorobeychik, Y., Luo, X., Liu, A. and Wang, T., 2020, October. A tale of evil twins: Adversarial inputs versus poisoned models. In Proceedings of the 2020 ACM SIGSAC Conference on Computer and Communications Security (pp. 85-99).

---

### Official Review · AnonReviewer4 · 2020-10-30
**Difficulties in learning using untrusted data.**

**Rating:** 7
**Confidence:** 4

**Review:**

This paper presents a very strong combined attack method, where infected
training examples are crafted such the the trojan backdoor becomes very
difficult to detect.  I feel their approach to be relevant, informative
and presents a significant advance.

The paper focuses on the mechanisms of cleverly disguising the Trojan training
data, and does an excellent evaluation.  The attack is particularly important
in some online training scenarios, where one might wish to use non-trusted
training data.

The evaluation is extensive, covering many existing and potential defenses,
with appendices covering different trojan types, trojan intensities, and attack
detectability, etc.  Readbility, and organization between main paper and
appendix material was good.

Given the strength of the attack, what practical implications does this have?
Appendix A addresses this for 2 cases that *fail* to defend, but a question I
found important was what steps would make the attack harder?  For example,
could App. A case (2) make the adversarial component more difficult if some
layers have parameters unknown, perhaps on a secure compute platform?
Does this have implications for how future compute platforms are set up?

p.7: Anomaly Index was actually *not* defined in appendix F, but in Wang et al.

This is a well-presented paper, with extensive experimental investigation
and should be published.

---

I was the only reviewer who happened to imagine their threat scenario had some importance.
After reviewing the authors' changes and comments, I feel that the threat scenario in the
revision still is insufficiently motivated/explained.  I'm downgrading to "good paper".

---

> ### Author Response · Authors · 2020-11-16
> **Responses to Reviewers Comments**
>
> We thank the reviewer for her/his feedback and constructive comments. In the following, we address the comments in order:
>
> [Point #1] We think that the reviewer’s comment on our use case analysis is interesting and insightful. We noticed that the specially designed, secure computing platforms (such as the Intel SGX) had been utilized in research for machine learning security-related topics. Taking this into consideration could definitely make the problem more interesting, challenging, and realistic. We thank the reviewer for providing such advice and will integrate it into our future works.
>
> [Point #2] The Anomaly Index is defined in the Wang et al. and we refer to it in Appendix F. We will revise the manuscript to resolve this confusing description.

---

> > ### Comment · AnonReviewer4 · 2020-11-16
> > **Perhaps consider implications for fine-tuning in future 5G networks**
> >
> > For [Point #1], I was specifically musing about a future 5G-ML environment. What implications does this attack have for online fine-tuning with public distributed data (say from cellphones)? What is the some [simplest?] future 5G-ML architecture that lessens the risk of the attack you describe?  Personally, I associate little risk with having Google/Microsoft/Amazon/company clouds running your attack in today's ML architectures.  In this respect I **agree** with other comments about the relevance of the attack.
> >
> > Instead, I see more relevance for future "IoT" scenarios that may argue *for* involving edge/fog compute when updating model parameters.

---

> > > ### Author Response · Authors · 2020-11-16
> > > **Implication of AdvTrojan for both centralized and decentralized settings**
> > >
> > > (1) We thank the reviewer for clarifying the comment. The “IoT” scenarios in the future 5G-ML environment are definitely relevant to AdvTrojan. When the scenarios involve edge/fog computing in a federated learning manner (Appendix A case 2), AdvTrojan becomes lethal and could be launched by one or more malicious participants, who can send bias gradients/parameters to the centralized server.
> > >
> > > (2) To lessen the risk of our attack, one potential solution is to create a validation set that includes sensitive data points to AdvTrojan along the decision boundary. If a participant sends its gradients/parameters, which often demonstrate poor performance over the validation set over time, we just ignore gradients/parameters from that participant in updating the centralized model.
> > >
> > > (3) In addition to the scenarios that involve edge/fog computing, AdvTrojan may also threaten centralized training scenarios, especially through model reusability, i.e., widely-used and practical platforms such as GitHub, Tekla, and Kaggle allowing users to upload and publish poisoned models.
> > >
> > > (4) Lastly, we agree with the reviewer that using trusted cloud-based platforms has little risk. However, outsourcing the model's training to (trusted and untrusted) cloud-based platforms, without a rigorous guarantee that the trained model is free of AdvTrojan, imposes a potential security risk. Even trusted cloud-based platforms (e.g., Google, Microsoft, Amazon, and company cloud) need to provide such a (liability) guarantee to the clients, in order to ease the (little) risk concern. This is a challenging task that has not been studied before.
> > >
> > > From (3) and (4), we believe that AdvTrojan is lethal even in centralized training manners.

---

### Official Review · AnonReviewer3 · 2020-11-02
**Simple combination of two known attacks**

**Rating:** 5
**Confidence:** 4

**Review:**

This paper presents a new attack against neural networks that combine Adversarial inputs and trojans. The key idea is to train a trojaned network that the victim might believe to be adversarially robust but the trojan will be activated when a trigger and adversarial noise are both present in the input image.

Strengths
-------------

- The attack presented by the authors beat most existing defenses

Weaknesses
-----------------
- The contribution seems very incremental given that there exist a substantial number of works in both adversarial inputs and trojaning. The core training algorithm (Alg. 1) also seems very straightforward.

- The threat model seems to be somewhat unrealistic. The adversarial inputs usually require norm-bounded global (all pixel) perturbations while inserting trojan triggers require local modifications to the input image. It is not clear to me what is achieved by putting these two different types of attacker models together that cannot be achieved by a regular trojan attack with larger triggers.

- the evaluation against existing trojan detection/prevention methods seem a bit unfair as they were never designed to consider adversarial inputs

Post Author response update
----------------------------------------
Based on the author's response, I will raise my score to 5.

---

> ### Author Response · Authors · 2020-11-16
> **Responses to Reviewers Comments**
>
> We thank the reviewer for her/his feedback. The following paragraphs address the points raised by the reviewer.
>
> [Point #1] Although both adversarial and Trojan attacks are studied in the literature, the idea of combining the attacks is new and important. Very recently, a research work combining the two attacks was published in the CCS 2020 [1]. Through careful design, our AdvTrojan combines the adversarial and Trojan attacks in a novel and new way compared to [1] such that the infected classifier could obtain two sets of behaviors (i.e., both robust and non-robust behaviors) that are controlled by the presence of the Trojan trigger. As a result, the model infected by our AdvTrojan can provide “fake robustness” that misleads the user to trust it as an adversarially trained model. In contrast, the combination attack proposed in [1] does not achieve this “fake robustness” goal, making our attack stealthier. We perform empirical analysis on toy examples to demonstrate how our attack can be activated. We also provided arguments on why existing defences cannot be modified to defend our attack. Mainly due to the fact that the search space becomes very large when we activate our attack. In addition to this novelty, our experiments also show that our AdvTrojan can bypass several state-of-the-art one-sided defenses (against adversarial and Trojan attacks). More importantly, our AdvTrojan can also break the defense proposed in [1], which aims at defending the combination attacks.
>
> [Point #2] By combining the vulnerabilities towards the adversarial perturbation and Trojan attack, the model infected by our AdvTrojan could obtain two sets of behaviors (i.e., both robust and non-robust behavior) that are controlled by the presence of the Trojan trigger. When the Trojan trigger is not contained in the input, the infected model works like an adversarially trained model. Otherwise, the infected model works like a vanilla model. As a result, the user, who does not know the Trojan trigger, is likely to be fooled and trust the infected model as an adversarially trained one. This property does not exist in models infected by large trojan inputs. Moreover, the novel design of AdvTrojan separates the attack into a two-step process (i.e., the Trojan trigger and the adversarial perturbation), which bypasses the state-of-the-art defenses for the adversarial attack, Trojan attack, and combination attack introduced in [1]. The research of combination attack is still in the early-stage and the one published in CCS 2020 presents the combination of adversarial and Trojan attacks. Compared with our AdvTrojan, the combination attack proposed in [1] does not achieve the “fake robustness”, making the attack even stealthier.
>
> [Point #3] We agree with the reviewer that the existing trojan detection/prevention methods are not considered adversarial inputs. Therefore, we also evaluated our AdvTrojan against defenses that try to combine the two attacks. One of these defenses is introduced in [1] for the combination attack. The other is designed by ourselves through modifying the existing Neural Cleanse defense introduced in [2]. Scarily, the results show that none of these defenses can defend our AdvTrojan. We believe this phenomenon reveals the importance of studying combination attacks like AdvTrojan.
>
> [1] Pang, R., Shen, H., Zhang, X., Ji, S., Vorobeychik, Y., Luo, X., Liu, A. and Wang, T., 2020, October. A tale of evil twins: Adversarial inputs versus poisoned models. In Proceedings of the 2020 ACM SIGSAC Conference on Computer and Communications Security (pp. 85-99).
> [2] Wang, B., Yao, Y., Shan, S., Li, H., Viswanath, B., Zheng, H. and Zhao, B.Y., 2019, May. Neural cleanse: Identifying and mitigating backdoor attacks in neural networks. In 2019 IEEE Symposium on Security and Privacy (SP) (pp. 707-723). IEEE.

---

### Decision · Program_Chairs · 2021-01-07
**Final Decision**

**Decision:**

Reject

**Comment:**

The paper presents a new attack combining trojans (backdoor attacks) with adversarial examples. The new attack is triggered only if both a trojan and the respective adversarial perturbation are present. Experimental evaluation demonstrates that neither adversarial training (as a defense against adversarial examples) nor defenses against backdoors are effective against the new attack.

The proposed method is original albeit somewhat incremental (combination of two well-known attack techniques). The main weakness of the paper, however, is its threat model. It is not clearly explained why the proposed attack would make sense for an attacker. Backdoor attacks are typically executed by model creators in order to force certain decisions on certain data. On the other hand, adversarial examples are generated by model users (or abusers) who have an interest in wrong model predictions (e.g., decisions made in their favor). The paper does not provide a convincing use-case in which such combined attacks would be feasible.

Furthermore, paper's clarity can be improved. The introduction does not present a clear picture of poisoning attacks. It essentially treats poisoning attacks as equivalent to backdoor/trojan attacks. This is not true and a substantial body of research (starting from the seminal paper by Barreno et al. in 2006) has addressed indiscriminate poisoning attacks aimed at general deterioration of classifier performance. A distinction between a clean-label and a poisoned-label attacks is also not clearly presented. The notation of Section 3 is rather complex and confusing.